# Heavy Metals in Sediment from the Urban and Rural Rivers in Harbin City, Northeast China

**DOI:** 10.3390/ijerph16224313

**Published:** 2019-11-06

**Authors:** Song Cui, Fuxiang Zhang, Peng Hu, Rupert Hough, Qiang Fu, Zulin Zhang, Lihui An, Yi-Fan Li, Kunyang Li, Dong Liu, Pengyu Chen

**Affiliations:** 1International Joint Research Center for Persistent Toxic Substances (IJRC-PTS), School of Water Conservancy and Civil Engineering, Northeast Agricultural University, Harbin 150030, China; ZhangFuxiang823@163.com (F.Z.); ijrc_pts_neau_paper@yahoo.com (Q.F.); kunyleee@163.com (K.L.); glwonder@163.com (D.L.); 18645148351@163.com (P.C.); 2State Key Laboratory of Simulation and Regulation of Water Cycle in River Basin, China Institute of Water Resources and Hydropower Research, Beijing 100038, China; hp5426@126.com; 3The James Hutton Institute, Craigiebuckler, Aberdeen AB15 8QH, UK; Rupert.Hough@hutton.ac.uk (R.H.); zulin.zhang@hutton.ac.uk (Z.Z.); 4State Environmental Protection Key Laboratory of Estuarine and Coastal Research, Chinese Research Academy of Environmental Sciences, Beijing 100012, China; anlhui@163.com; 5IJRC-PTS, State Key Laboratory of Urban Water Resource and Environment, Harbin Institute of Technology, Harbin 150090, China; ijrc_pts_hit06@yahoo.com

**Keywords:** heavy metals, sediment, contamination characteristics, possible source, ecological risk

## Abstract

The concentrations and ecological risk of six widespread heavy metals (Cu, Cr, Ni, Zn, Cd and Pb) were investigated and evaluated in sediments from both urban and rural rivers in a northeast city of China. The decreasing trend of the average concentration of heavy metals was Zn > Cr > Cu > Pb > Ni > Cd in Majiagou River (urban) and was Zn > Cr > Pb > Cu > Ni > Cd in Yunliang River (rural). The results showed that the concentrations of Cd and Zn were significantly elevated compared to the environmental background value (*p* < 0.05). Half of all sampling locations were deemed ‘contaminated’ as defined by the improved Nemerow pollution index (*P_N_’* > 1.0). Applying the potential ecological risk index (*RI*) indicated a ‘high ecological risk’ for both rivers, with Cd accounting for more than 80% in both cases. Source apportionment indicated a significant correlation between Cd and Zn in sediments (*R* = 0.997, *p* < 0.01) in Yunliang River, suggesting that agricultural activities could be the major sources. Conversely, industrial production, coal burning, natural sources and traffic emissions are likely to be the main pollution sources for heavy metals in Majiagou River. This study has improved our understanding of how human activities, industrial production, and agricultural production influence heavy metal pollution in urban and rural rivers, and it provides a further weight of evidence for the linkages between different pollutants and resulting levels of heavy metals in riverine sediments.

## 1. Introduction

Heavy metal pollution in the aquatic environment has attracted extensive concern due to its environmental persistence, potential adverse effects on human health and accumulation in the food chain [1,2]. Once heavy metals enter the river, depending on the physico-chemical characteristics of the river, they may be adsorbed to suspended particulate matter and later deposited to the sediments under the action of gravity [3]. Thus, riverine sediments often act as a sink for heavy metals, leading to elevated concentrations in sediments compared to inputs into the riverine system. If hydrodynamic conditions change or if changes to physico-chemical equilibria occur, metals present in the sediments can be re-released into the water, thus causing secondary pollution [4]. Therefore, where sediments act as a “sink” or “secondary source” for heavy metals, there is potential to use the sediments as an effective environmental medium to monitor and evaluate the magnitude and sources of heavy metal pollution in the aquatic environment [5,6,7].

With the rapid development of industry, the regular/increasing use of pesticides and fertilizers, and the increasing intensity of human activities, large volumes of wastewater containing heavy metals are discharged into aquatic systems. The atmospheric deposition of heavy metals from aeolian sources could also lead to high pollution levels in water and sediment [8]. Pollution with heavy metals also has the potential to occur during the processing and use of fossil fuels [9]. Thus, water pollution has become an important issue that influences ecological quality and the sustainable development of the social economy.

In China, the contamination of heavy metals in sediment from Pearl River, Liao River, Yangtze River and Songhua River has caused widespread concerns since the late 1980s [10,11,12,13,14]. The statistical evaluation of Cao [15] indicated that there was an increasing trend of heavy metal pollution from the north to south of China. Additionally, the concentrations of heavy metals in sediments have generally been found to be elevated in urban rivers compared to suburban and rural rivers [16], but this urban-rural/suburban spatial distribution pattern might be diffused with urbanization [17]. Due to these concerns, various indices and tools have been established for identifying potential the ecological risk from heavy metal pollution as well as to support subsequent management/mitigation—these include the Nemerow pollution index [18], the geo-accumulation index (*I_geo_*) [19,20], and potential ecological risk [21]. Though the traditional Nemerow pollution index has been widely used to assess ecological risk, it has a tendency to over-estimate ecological risk because it adopts a very precautionary approach to risk estimation.

Harbin, one of the most important equipment manufacture and food production bases in China, straddles the Songhua River. More than ten rivers flow through the city of Harbin, of which Majiagou and Yunliang River are two representative rivers flowing through urban and rural areas, respectively. As a result of rapid industrialization (Majiagou catchment) and agricultural intensification (Yunliang catchment), both rivers have a relatively long history of receiving inputs/discharges of a large range of pollutants. However, there have been few comprehensive comparative studies on the distribution and sources of heavy metals, as well as the associated ecological risks for urban vs. rural rivers. Thus, the objectives of this study were: (1) to reveal contamination levels and spatial distribution characteristics of heavy metals in the sediments of the Majiagou River and Yunliang River; (2) to identify the possible sources of heavy metals by Pearson’s coefficient coupled principle component analysis (PCA); and (3) to evaluate the ecological risk by using the improved Nemerow pollution index and the potential ecological risk index.

## 2. Materials and Methods

### 2.1. Study Area

The study area has a temperate continental monsoon climate [22]; the average temperature is −19 °C in January and 23 °C in July, and the annual average temperature is 3.5 °C. Rainfall is mainly concentrated in the period from June to August. The Majiagou River (126°41′–126°43′E and 45°32′–45°49′N) flows through the central area of the city, and its course can be divided into three sections which include the urban section (mUR, M1–M5), industrial zone (mIZ, M6–M8) and suburban section (mSU, M9–M12). By contrast, the catchment of the Yunliang River (126°17′–126°38′ E and 45°30′–45°41′ N) is dominated by agricultural production, and thus these sampling sites named rural section (yRU, Y1–Y6). Both these rivers are important tributaries of the Songhua River, which is the major source of drinking water for inhabitants of Harbin and irrigation water for one of the most important food production bases in China [23]. Detailed information on sampling sites is illustrated in Figure 1.

### 2.2. Sample Collection

A total of 18 surface sediment samples (12 samples (M1–M12) in Majiagou River and 6 samples (Y1–Y6) in Yunliang River) were collected in October 2017. Sediment was collected by grab sampling (0–10 cm from the surface) and stored in brown glass bottles that had been pre-washed with nitric acid. At each sampling location, three samples were taken 30 m apart, mixed well, and then pooled to produce one representative sample per site. All sediment samples were stored in a cooled container and transported to the International Joint Research Center for Persistent Toxic Substances (IJRC-PTS) laboratory at Northeast Agricultural University (Harbin) as soon as possible, and they were then stored in a refrigerator prior to digestion.

### 2.3. Sample Processing and Analysis

The treatment of the sediment sample was similar to the procedures used for the determination of heavy metals in the certified reference material for the environmental quality standard for soils (GB15618-1995) [24]. The sediment samples were lyophilized, and plant roots, gravel and other foreign matter were removed prior to grinding. Approximately 0.5 g of ground sample was digested in a Teflon crucible on a hot plate by wet digestion (HCL–HNO_3_–HClO_4_–HF) (guaranteed reagent, Tianjin Yaohua Chemical Reagent Co., Ltd.), until there were no obvious solid particles in the crucible and no white smoke escaped. At this point, the crucible was removed from the hot plate and allowed to cool to room temperature. The digestate was then diluted to 50 mL using deionized water, and it was mixed thoroughly before storage at 4 °C prior to instrumental analysis. The concentrations of heavy metals in the pretreated samples were determined using the ICE 3500 (Thermo Fisher Scientific, Waltham, MA, USA) atomic absorption spectrophotometer; Cu, Cr, Ni, Zn were measured using the flame portion, and the graphite furnace portion was used for the detection of Cd and Pb.

### 2.4. Quality Assurance/Quality Control

All the crucibles and glass containers were soaked in 10% HNO_3_ for 24 h, washed with ultrapure water, and dried before use. Blank and standard samples were used—one per each set of 6 sediment samples. The equivalent acid was added into a Teflon crucible without any exogenous substances in it, and the same digestion program was performed with the sediment sample to make the blank sample. Standard reference materials (GBW07305) from the Chinese Academy of Measurement Sciences were used to make the standard sample, and then they were digested and analyzed using the same procedure. The average recovery rate was 93%, and the concentration of heavy metals in blank samples was always below the minimum detection limits. The correlation coefficient of calibration curves of the 6 heavy metals was greater than 0.9995 (0.995 is the minimum permissible limit for instrument test), and the standard deviation between parallel samples was less than 5%.

### 2.5. Pollution Assessment Methods

#### 2.5.1. Single Factor Pollution Index and Improved Nemerow Pollution Index

The single factor pollution index (*P_i_*) can be used to assess the magnitude of pollution attributed to single pollutants in sediment. Deriving *P_i_* for each measured pollutant in turn can be useful for highlighting the most important pollutant in the suite of pollutants investigated [6]. The single factor pollution index for heavy metals is calculated as:(1)$$P_{i}=C_{i}/C_{iref}$$where *C_i_* is the measured concentration of heavy metals and *C_iref_* is the environmental background value which represents the element content of environment medium in the case of without any influences by exogenous substances. Here we chose the I standard value of the Environmental Quality Standard for Soils (GB15618-1995) proposed by the State Environmental Protection Administration of China (SEPA) (Cu: 35 mg/kg, Cr: 90 mg/kg, Zn: 100mg/kg, Pb: 35mg/kg, Ni: 26 mg/kg and Cd: 0.2 mg/kg) [24].

The Nemerow pollution index [18] has been widely applied in the evaluation of heavy metal pollution. However, this method has a tendency to over-estimate the magnitude of heavy metal pollution [25]. This is because the method neglects differences in the toxicological profiles of the different metals as well as their relative importance. Thus, the Nemerow pollution index can be modified using different weighting factors that act as proxy measures for the biological toxicity and relative importance of the different heavy metals.

In this study, the weighting factors were derived using the method of Deng [25]. Briefly, a comprehensive weight was derived from the relative importance of each heavy metal (*R^r^_i_* = *C_imax_*/*C_iref_*) and the relative toxic importance (*R^t^_i_ = T_imax_*/*T_iref_)*; where *C_imax_* and *T_imax_* are the maximum background concentrations and maximum toxicity for each heavy metal, respectively, and *T_iref_* refers to the toxicity coefficient (Cd = 30, Cr = 2, Zn = 1, Cu = Pb = Ni = 5.) [21,26]. The comprehensive weight was calculated by:(2)$$w_{i}=\frac{R_{i}^{r}}{2\sum_{i=1}^{n}R_{i}^{r}}+\frac{R_{i}^{t}}{2\sum_{i=1}^{n}R_{i}^{t}}$$

The equations for calculating the traditional Nemerow pollution index (*P_N_*) and improved the Nemerow pollution index (*P_N_’*) are as follows:(3)$$P_{N}=\sqrt{\frac{P_{iave}^{2}+P_{i}{}_{\max}^{2}}{2}}$$
(4)$$P_{N}^{\prime }=\sqrt{\frac{P_{iave}^{2}+P_{i}{}_{w_{i}\max}^{2}}{2}}$$
where *P_N_* is the improved Nemerow pollution index; *P_iave_* and *P_imax_* are the mean and maximum values of the single pollution index, respectively; and *P_iwmax_* is the top pollution factors of comprehensive weight in all the pollution factors.

On this basis, this study also determined the corresponding evaluation criteria according to environmental quality standard for soils (GB15618-1995) [24] in order to better reflect the comprehensive effect of heavy metal pollution objectively (Appendix A).

#### 2.5.2. Potential Ecological Risk Index

The potential ecological risk index (*RI*) was employed to assess the potential risks of one or multiple ecological factors [21]. The classification standard of *RI* proposed by Hakanson was based on considering the toxicity of 8 parameters (polychlorinated biphenyls (PCBs), Hg, Cd, As, Pb, Cu, Cr and Zn). In this study, only 6 of these parameters were included, so the classification was adjusted using the method of Chen [27]. The potential ecological risk index (*RI*) was calculated as follows:(5)$$RI=\sum_{i=1}^{m}E_{r}^{i}$$
(6)$$E_{r}^{i}=T_{iref}\times P_{ji}$$
where *RI* is the potential ecological risk index, *E_r_^i^* is the single ecological risk index of each heavy metal, *P_ji_* is the single pollution index, and *T_iref_* is the toxicity coefficient of each heavy metal [21,26].

The classification of *P_i_*, *P_N_*, *E_r_* and *RI* are presented in Appendix A.

#### 2.5.3. Statistical Analysis

A Pearson correlation matrix (PCM) and principal component analysis (PCA) were performed in order to elucidate any associations between the heavy metals. Prior to this, the Kolmogorov–Smirnov test was used to confirm normality. One-sample and independent-samples t-tests were conducted to identify within-river and between-river differences in concentrations of the heavy metals in sediments. A Pearson correlation analysis was then applied to analyze the strength of association between the detected contents of heavy metals, which was considered to be significant if *p* < 0.05.

A principal component analysis (PCA) was used to help identify the potential sources of contamination. Results were refined by applying a varimax rotation in order to reduce the number of heavy metals that have high loadings on each factor [28]. Prior to the PCA, through Kolmogorov–Smirnov test identified that all the heavy metal data were fit for normal distribution (*p* > 0.05), the suitability of the data for factor analysis was determined using the Kaiser–Meyer–Olkin (KMO) test for sampling adequacy as well as the Bartlett sphericity test. Additionally, all the data (a total of 72) were qualified in which the KMO > 0.5 and the significance level of the Bartlett sphericity test *p* < 0.001. In this study, factors with a cumulative contribution of variance >90% were selected for inclusion.

## 3. Results

### 3.1. Concentrations

The concentrations of six heavy metals in the surface sediments of Majiagou River and Yunliang River are presented in Figure 2. The concentrations of measured heavy metals in Majiagou River were: Cu (4.00–82.54), Cr (75.12–203.15), Zn (128.17–1416.71), Pb (8.86–57.49), Ni (7.91–30.38), and Cd (0.08–4.08) mg/kg dw (dry weight). The average concentrations of heavy metals decreased in the following order: Zn (358.54) > Cr (107.37) > Cu (28.05)> Pb (26.98) > Ni (17.82) > Cd (0.76) mg/kg. Overall, the concentration of Zn was significantly higher than the environmental background value (*p* < 0.05), while the Ni concentration was much lower than background (*p* < 0.01). However, there were no similar differences observed for the other four metals (*p* > 0.05), indicating that these were not elevated above background concentrations. The average and concentration ranges of heavy metals in the Yunliang River were: 19.46 (15.75–22.29) for Cu, 68.19 (53.65–81.92) for Cr, 861.63 (113.23–2474.05) for Zn, 32.75 (9.31–114.42) for Pb, 8.16 (Below the detection limit (BDL)–13.11) for Ni, and 1.83 (BDL–4.29) mg/kg for Cd, respectively. The average concentrations of Cd (1.83 mg/kg) and Zn (861.63 mg/kg) were 9.15 and 8.62 times higher than their environmental background values (0.2 mg/kg for Cd and 100 mg/kg for Zn), respectively. The measured concentrations of heavy metals in sediments from the Majiagou River and Yunliang River were compared with those found in other studies (Appendix A). The mean concentrations of all heavy metals measured in this study (except for Zn) were significantly lower than those in the Xiangjiang River (*p* < 0.01), which is one of the most polluted rivers in China [29,30]. The concentrations of Pb and Ni measured in this study were lower than those detected in the Louro River in Spain (*p* < 0.01) [31], the Gorges River in Australia (*p* < 0.05) [32] and the Gironde Estuary in France (*p* < 0.01) [33], all of which are heavily polluted. The concentrations of Cd and Zn in the Majiagou and Yuliang Rivers were found to be greater than many of the Chinese rivers included in Appendix A (*p* < 0.05). For example, the mean concentrations of Cd and Zn measured in the studied rivers were about 4–20 times higher than those in the Yangtze River [34] and Yellow River [6]. In addition, the average concentrations of Cr in the sediment from the Majiagou River was similar to measurements reported from the Songhua River, which tends to have elevated levels of Cr compared to other Chinese rivers [35]. Emissions from coal combustion, especially during winter, could lead to a high concentration of Cr in sediments in the study area.

### 3.2. Spatial Distribution

The concentrations of heavy metals in sediments of both rivers appear to be a function of land use and its spatial distribution (Figure 3). As expected, Cr and Ni measured in sediments from industrialized parts of the Majiagou River catchment were elevated compared to rural sections (*p* < 0.05). The highest concentration of Cu and Pb occurred in sediments from the urbanized areas of the Majiagou River catchment, while the maximum concentration of Zn was in Yunliang River. The average concentrations of heavy metals in the Yunliang River were higher than those in the suburban section of Majiagou River except for Cr and Ni. Figure 3 suggests that Ni has a relatively lower degree of dispersion within this study area, which may indicate that the majority of the Ni is derived from geogenic sources rather than from human activities. Overall, the concentrations of heavy metals were greatest in sediments located within the urban and industrial areas of Majiagou River compared to the suburban areas of Majiagou River and the Yunliang River. The only exception to this trend was Zn, which was more abundant in sediments from the Yunliang River, suggesting that runoff from agricultural production could be an important source [36,37].

### 3.3. Possible sources

Inferences regarding the possible sources of heavy metals in sediments of the Majiagou River and Yunliang River were developed using the Pearson correlation coefficients within- and between heavy metals measured at all sampling sites. There was a significant correlation between Cd and Zn in the sediments of the Yunliang River (*R* = 0.997, *p* < 0.01) (Appendix A), indicating that they could have similar sources. The coexistence of Cd and Zn within an agricultural catchment suggests agronomic sources such as the excessive use of phosphate fertilizer and pesticides, which can enter the river via soil runoff [38,39]. There was a significant correlation between Ni and Cr in the sediments of the Majiagou River (*R* = 0.74, *p* < 0.01) (Appendix A). Additionally, Pb has a significant correlation with Zn (*R* = 0.79, *p* < 0.01) and Cd (*R* = 0.73, *p* < 0.01), respectively (Appendix A). These results indicate that the sediments of the Majiagou River could be receiving multiple pollutants from the same emission sources or at least spatially-similar sources.

While a Pearson correlation analysis (Appendix A) can be used to make inferences about sources for the heavy metals, it is a relatively simplistic analysis given the complexity of the riverine environment. Therefore, a PCA was also applied to the data from Majiagou River areas, because its flow patterns, and hence its dispersion of metals, are known to be particularly complex. As illustrated in Figure 4, the first principal component (PC1) accounted for 58.6% of the total variance and was heavily associated with Zn, Cd and Pb (consistent with the Pearson’s correlation analysis; Appendix A). The PC1 could originate from industrial activities because Harbin is an important industrial base in Northeast China with a long history of equipment manufacturing. It has been reported that Zn in urban settings is mainly derived from the sewage discharge from chemical enterprises, the processing of Zn containing minerals, the manufacture of metal machinery, and the wear and tear of automobile tires [40]. Cd is likely to be from electronics, printing and dyeing, electroplating and chemical industry sources [41]. Pb tends to originate from the industrial utilization of minerals containing lead and the combustion of fossil fuels. All these sources are therefore likely to be present within the industrialized areas of Harbin.

PC2 accounted for 22.8% of the total variance, is highly loaded with Cr and Ni, and corroborates the Pearson’s correlation analysis between Cr and Ni (Appendix A). Mineral weathering and atmospheric deposition from coal-burning dust could lead to the accumulation of Ni and Cr in sediments [39,42,43]. Thus, one inference is that PC2 may represent a combination of coal combustion and natural sources.

PC3 accounted for 14.2% of the total variance and is highly loaded with Cu. This may have been caused by the emissions of the vehicle exhaust and brake pad wear [44,45], while the high enrichment of Cu in the soil along the main street of Harbin City has been investigated and thought attributable to traffic sources [46]. Thus, we infer that PC3 originated from traffic sources.

### 3.4. Pollution Degree Assessment

The spatial distribution of *P_i_* in the sediments of the Majiagou River and Yunliang River is presented in Figure 5**,** with the values of *P_i_* following a decreasing trend of Cd > Zn > Cr > Cu > Pb > Ni. Over one third of sampling sites were assigned ‘high’ pollution status on the basis of their contents of Cd and Zn alone. The average *P_i_* value of Cr was 1.19, indicating that the levels of Cr pollution were ‘low.’ The coefficient of variation for Cr was 0.32, which corresponds to a moderate variability, suggesting that the sources for Cr are more likely to be diffuse pollution associated with atmospheric deposition, agricultural activities, and discharge from industrial and domestic wastewater. However, the average *P_i_* values of Cu, Pb and Ni were less than 1, indicating that these areas are relatively less polluted. It can be seen from Figure 5 that the *P_i_* values in sediments of Yunliang River tended to be lower compared to those from the Majiagou River. The exceptions to this observation re Cd and Zn, which have average *P_i_* values of 6.11 and 8.61, respectively, indicating ‘high’ levels of pollution (as defined in Appendix A). Levels of Ni in both rivers could be considered ‘clean.

The improved *P_N_’* estimated that 58% of all sampling sites in the sediment of the Majiagou River were polluted by heavy metals (Figure 6), of which the M7 and M8 sampling sites within the industrial area were considered to have ‘moderate’ and ‘high’ levels of pollution (according to Appendix A), respectively. In addition, about 60% of the sampling sites categorized as having ‘moderate pollution’ were located within the urban area of the Majiagou River, while *P_N_’* defined the suburban area as ‘clean.’ Thus, emissions and discharges from industrial production need further attention from government, public, and other stakeholders because this study suggests that they might be the most important contributors to heavy metals in the riverine environment. Compared to the Majiagou River, the majority of sampling sites along the Yunliang River indicated ‘no pollution.’ The exception to this were the Y1 and Y3 sites, where *P_N_’* values of 10.6 and 16.3 (‘serious pollution’), respectively, were determined. This might indicate point sources of pollution at these two locations.

### 3.5. Potential Ecological Risk Assessment

The *RI* was employed to quantitatively evaluate the ecological risk level of heavy metals in the sediments of the Majiagou River and Yuliang River. The values of *E_r_^i^* and *RI* of each sampling site according to Equations (5) and (6) are illustrated in Figure 7.

Due to its high relative toxicity, about 80% of the potential ecological risk posed by heavy metal contamination in sediments of the two rivers could be attributed to Cd (Appendix A). According to the results of *E_r_*, about 50% of values for Cd were greater than 40 (‘moderate’ ecological risk, or higher (Appendix A)). Ecological risks associated with Cd were especially pronounced at M8 (*E_r_* = 612.7), Y1 (*E_r_* = 418.5) and Y3 (*E_r_* = 644), which are defined as ‘serious ecological risk’ according to Appendix A. Considerably lower ecological risks were associated with the other five heavy metals (Zn, Pb, Ni, Cr and Cu).

The mean values of *RI* were 130.41 and 201.91 in sediments of the Majiagou River and Yunliang River, respectively, which is indicative of ‘high ecological risk’ (Appendix A). The discharge from industrial and domestic wastewater might be the primary driver of the ecological risk in sediments of the Majiagou River. Despite its rural catchment, the elevated levels of Cd in sediments of the Yunliang River enhanced the *RI* value, especially at the Y1 and Y3 sites. Given that riverine sediments can act as both a sink and source of heavy metals, sites such as Y1 and Y3 have the potential to be implicated in the future re-release of Cd into the aquatic environment and any associated consequences. This could involve food-chain related exposure and potential human health risks due to the exploitation of the Yunliang for irrigation water for agricultural production.

## 4. Discussion

Though the single factor pollution index method has been widely used, it is only applicable to a single pollutant and does not take into consideration the mixture of heavy metals often present in pollution situations. While, the improved *P_N_’* as a multiple element index integrates the average value of the pollution index (*P_iave_*) for individual sites and the single pollution index (*P_iwmax_*) (Equations (2)–(4)). The improved values (*P_N_’*) were lower than traditional *P_N_*; this was especially apparent in the Yunliang River where values of *P_N_* were almost three times as *P_N_’* (except for Y1 and Y3), which therefore resulted in a different conclusion when determining the degree of pollution (Figure 6). These differences can be attributed to the influence of overemphasis on the maximum pollution factors on the final results in the derivation of *P_N_*. For the Y4, Y5 and Y6 sampling sites, the maximum pollution factors (Zn) were more than 2.6, 6.1, and 2.5 times greater than the other heavy metals, respectively, and, more importantly, there was only one factor (Zn) that was considered to be ‘moderate’ pollution, while the others were considered ‘clean,’ including the top factor of weight (Cd). This phenomenon was also found, although less pronounced, at the other sites. Comparatively, the improved Nemerow index provided a less bias evaluation of the quality of sediments by taking full consideration of the relative importance and biological toxicity of heavy metals.

Both of the Majiagou River and Yunliang River are important tributaries of the Songhua River, and thus their water environment quality will affect the security of drinking and irrigation water for inhabitants and agricultural production along the Songhua River. Thus, different regulatory measures should be paid to the environment treatment in the future for these two rivers according to the correspondent pollution characteristics. Optimization and control in agricultural management might be the adapted scheme for reducing the input of pollution sources in Yunliang River, where the most important sources appear to mainly be from agricultural activities. However, the industrial areas, located in the middle and upper reaches of Majiagou River, appear to be the priority to control and management for reducing the input of pollutants from the wastewater discharge and atmospheric deposition, as well as avoiding the adverse influence on population density areas in the lower reach.

## 5. Conclusions

The concentrations, the possible sources and ecological risk of six heavy metals in sediments from urban and rural rivers were investigated in Harbin. The results showed that the concentrations of heavy metals in the urban and industrial areas of the Majiagou River were significantly elevated compared to those measured in sediments from suburban and rural areas. The exception to this was Zn, with the highest concentrations measured in sediments from the predominantly rural Yunliang River. It is possible that the excessive use of fertilizers and pesticides could be responsible for the elevated levels of Cd and Zn measured in the Yunliang River sediments, given the land use of this catchment is dominated by crop production. The source apportionment by Pearson correlation coupled with the PCA indicated diverse sources in the sediments of the Majiagou River, with Zn, Cd and Pb being thought to originate from industrial activities, Ni and Cr thought to be mainly derived from coal combustion and natural sources, and Cu thought to be mainly from traffic emissions. However, it must be noted that this is not a formal source apportionment and is reliant on inferences drawn from the available information. The improved Nemerow pollution index indicated a higher incidence and magnitude of pollution in the Majiagou River compared to the Yunliang River, and this was most acute in the urban and industrial parts of the catchment. The potential ecological risk assessment indicated high ecological risks associated with the sediments of both rivers, of which the *E_r_* of Cd was significantly higher than the other metals (Cd accounted for more than 80% of the *RI*; *p* < 0.01). Given the fact that riverine sediments can act as both a sink and a source for heavy metals, there is potential for Cd to be implicated in secondary pollution events that could have wide implications, e.g., when river water is used to irrigate food crops.

## Figures and Tables

**Figure 1 ijerph-16-04313-f001:**
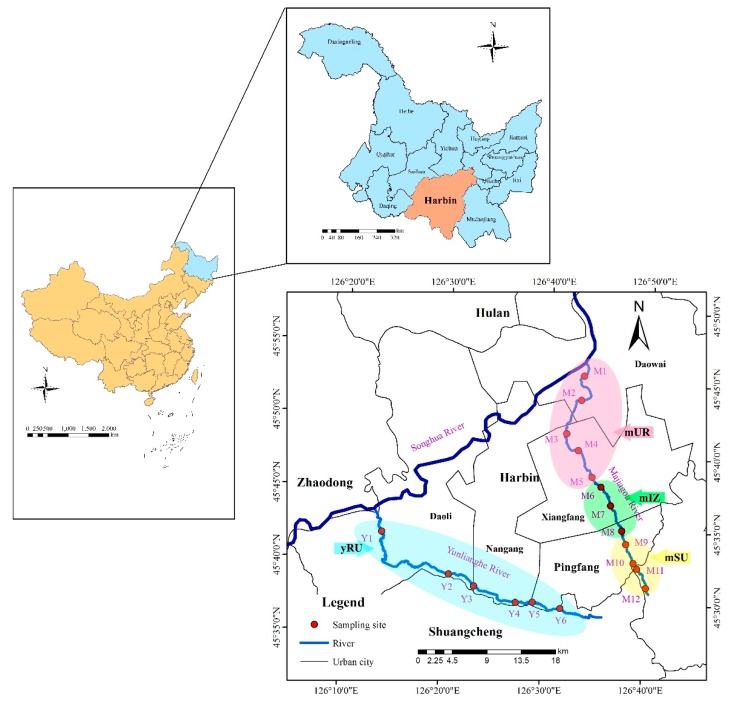
Locations of sampling sites in Majiagou River (M1–M12) and Yunliang River (Y1–Y6) in Harbin City.

**Figure 2 ijerph-16-04313-f002:**
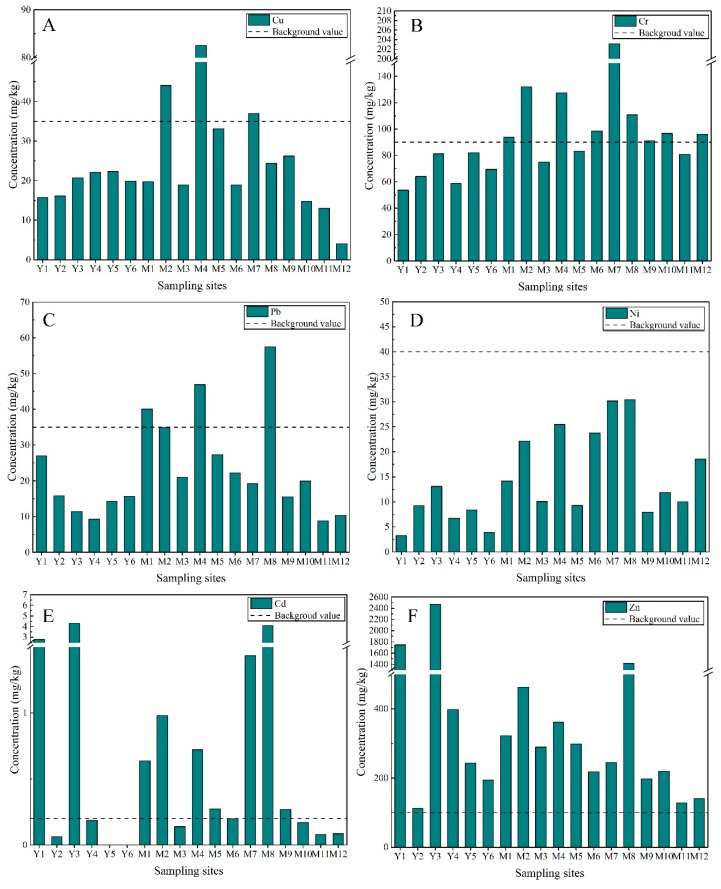
Concentrations of Cu (**A**), Cr (**B**), Pb (**C**), Ni (**D**), Cd (**E**) and Zn (**F**) in surface sediments of Majiagou River and Yunliang River in Harbin City (mg/kg).

**Figure 3 ijerph-16-04313-f003:**
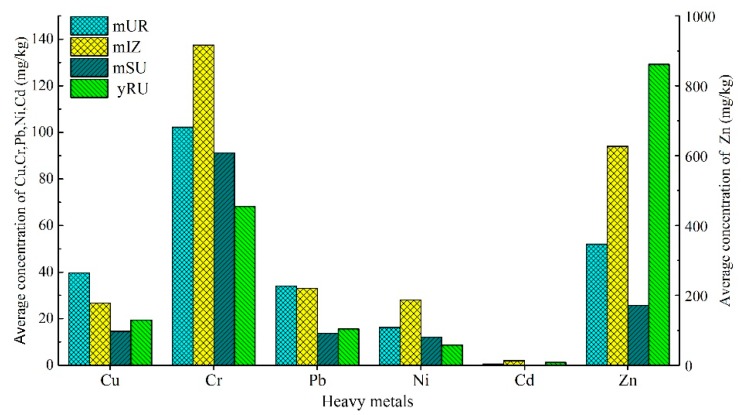
Average concentration of Cu, Cr, Pb, Ni, Cd and Zn in sediment of Majiagou River and Yunliang River at different functional areas.

**Figure 4 ijerph-16-04313-f004:**
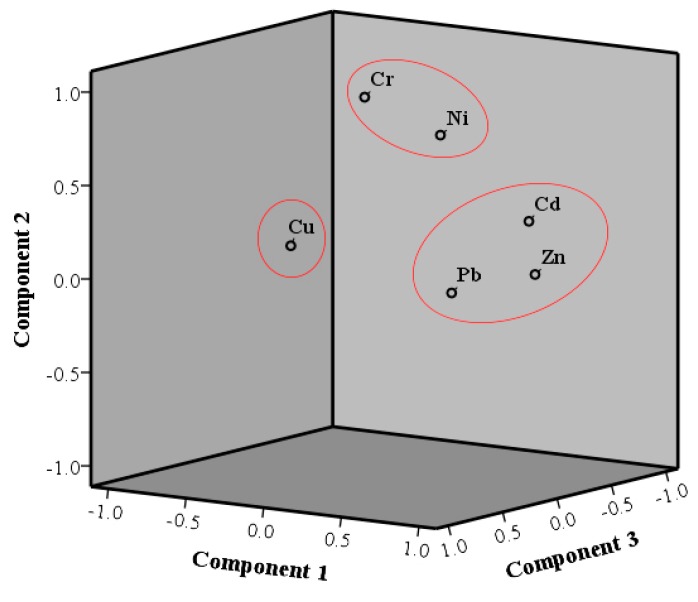
Plot of loading of three principle components.

**Figure 5 ijerph-16-04313-f005:**
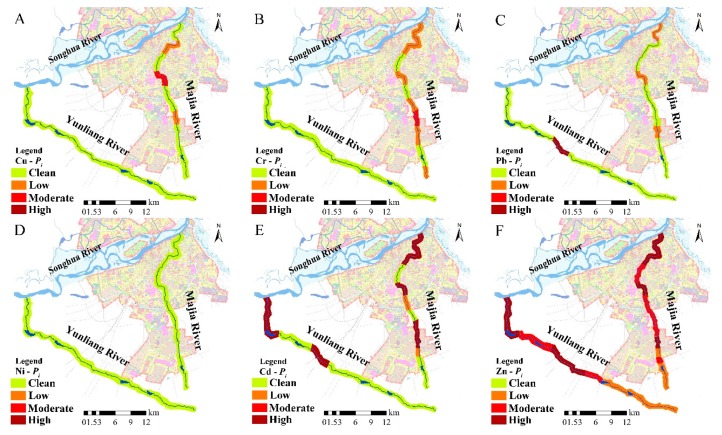
Spatial distribution of Cu (**A**), Cr (**B**), Pb (**C**), Ni (**D**), Cd (**E**) and Zn (**F**) in surface sediments by the single factor pollution index (*P_i_*).

**Figure 6 ijerph-16-04313-f006:**
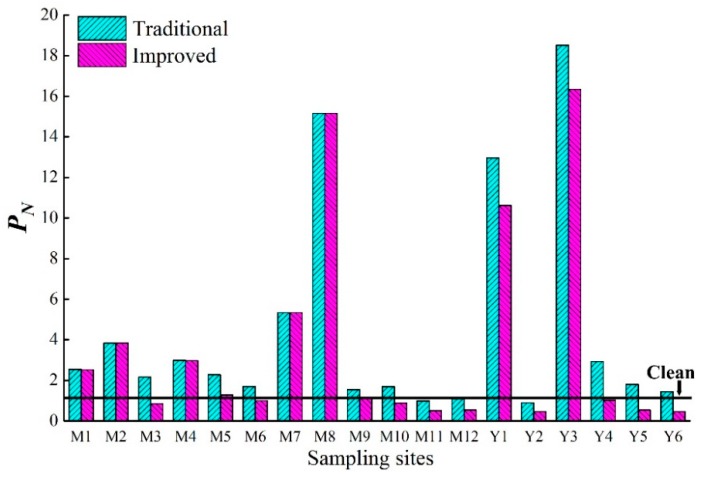
The Nemerow pollution index of each sampling site.

**Figure 7 ijerph-16-04313-f007:**
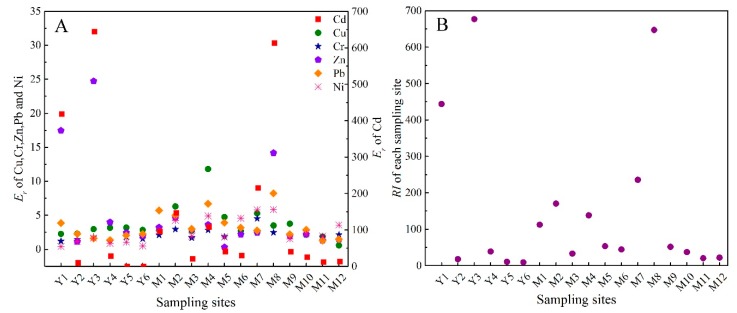
The value of the single ecological risk index (**A**) and the comprehensive potential ecological risk index (**B**) at each sampling site.

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
