# Peer review of "Heavy Metals in Sediment from the Urban and Rural Rivers in Harbin City, Northeast China"

_ijerph, 2019, doi:10.3390/ijerph16224313_

Round 1
Reviewer 1 Report
I find that the manuscript titled: "Heavy metals in sediment from the urban and rural rivers in Harbin City, Northeast China", is a very interesting paper. Moreover, this paper was trying to study the concentrations and presence of the ecological risk of six widespread heavy metals (Cu, Cr, Ni, Zn, Cd and Pb) in sediments from both urban and rural rivers Majiagou and Yunliang in a northeast city of China. This pollution process is very actual problem today worldwide in different areas.
In this study by applying the potential ecological risk index (RI) was indicated ‘high ecological risk’ for both rivers. In addition, the source apportionment indicated a significant correlation between Cd and Zn in sediments in Yunliang River, suggesting that agricultural activities could be the major sources. On the other hand, the industrial production, coal burning, natural sources and traffic emissions are indicated as main pollution sources for heavy metals in Majiagou River. However, very important is that both mentioned rivers are important tributaries of the Songhua River, which is the major source of drinking water for inhabitants of Harbin, and irrigation water for one of the most important food production bases in China.
In this regard, no specific proposal for remediation, control and stopping of the further pollution has been presented. Could you please clarify these details?
I recommend acceptance for publication after minor modifications.
Reviewer 2 Report
on page 3, lines 96-97 and 107 as well as on page 4 line 116 where different types of acids are mentioned, I think it is necessary to indicate the percentage of acid, the manufacturer, the degree of purity - essentially because the acids contain a certain concentration of heavy metals, so that according to her certificate possible pollution from it can be seen on page 4, line 112 when referring to the device and the calibration, it would be well to indicate which calibration standards were used on page, line 128 where reference value of each element is mentioned, it would be good to state according to which guidelines and for what type of water this applies why does the References list on supplementary file goes to number 8, and in the text the reference mark goes to number [47] on Figure 5, the river pollution color on the map is quite difficult to recognize, especially on the Majiagou River when it flows through an urban area. I think the color of the city should be removed so that the color of the river comes to the fore.
Reviewer 3 Report
Heavy metals in sediment form the urban and rural rivers in Harbin City, Northeast China.
Song Cui et al.2019. International Journal of Environmental Research and Public Health.
This manuscript reports on a study characterizing six heavy metal concentrations in sediments of two rivers in Harbin as a function of land use and spatial distribution, and their respective ecological risk. The study has clearly stated objectives and research questions, and demonstrates a good scientific merit, is well designed, and the authors provided good data interpretation. The manuscript is well structured and well written. Figure 5 shows an elegant way of presenting pollution results on a map. I recommend this manuscript for publishing with some minor, but necessary corrections.
My concerns relate to missing information, statistical details and explanation, and some spelling errors as follows:
Line 104: “…for the Environmental quality standard…”, suggest environmental lower case.
Line 117: “Blank and standard samples,…”, you need to explain what material was used for blanks and what are standard samples in your study, whether it is soil, sand, silica or something else, how it was spiked, with what and at what concentration?, please provide details.
Line 150: “(Table S1)”, S mentioned for the first time, so suggest (Table S1 in Supplementary Information).
Line 167: Prior to PCA, you need to specify sample size (how many samples were accepted for multivariate analysis and how many rejected), and if the sample size is small, you need to discuss the impact on eigenvalues.
Lines 181, 188: you discuss the concentration of metals being higher or lower than “the environmental background value”, but you have never described the background value definition. It is also used in Figure 2, but without explanation what are and how were derived background values it is meaningless. In the Materials and Method section, please define and describe what are “the environmental background values”, and how the background values were derived. This information is essential and very important.
Line 302: “…Figure 6. The nemerow pollution…”, suggested: The Nemerow pollution… .
